# Impact of AI Assistance in Pneumothorax Detection on Chest Radiographs Among Readers of Varying Experience

**DOI:** 10.3390/diagnostics15202639

**Published:** 2025-10-19

**Authors:** Chen-Wei Ho, Yu-Lun Wu, Yi-Chun Chen, Yu-Jeng Ju, Ming-Ting Wu

**Affiliations:** 1Department of Radiology, Kaohsiung Veterans General Hospital, No. 386, Dazhong 1st Rd., Zuoying Dist., Kaohsiung City 813414, Taiwan; sumei4152@gmail.com (C.-W.H.); jumpjump612@gmail.com (Y.-C.C.); wu.mingting@gmail.com (M.-T.W.); 2Department of Radiation Oncology, Kaohsiung Veterans General Hospital, No. 386, Dazhong 1st Rd., Zuoying Dist., Kaohsiung City 813414, Taiwan; seansaint27@gmail.com; 3School of Occupational Therapy, College of Medicine, National Taiwan University, No. 1, Sec. 1, Ren’ai Rd., Zhongzheng Dist., Taipei City 100233, Taiwan; juyujeng@gmail.com; 4School of Medicine, College of Medicine, National Yang Ming Chiao Tung University, No. 155, Sec. 2, Linong St., Beitou Dist., Taipei City 112304, Taiwan; 5Institute of Clinical Medicine, College of Medicine, National Yang Ming Chiao Tung University, No. 155, Sec. 2, Linong St., Beitou Dist., Taipei City 112304, Taiwan

**Keywords:** artificial intelligence, radiography, pneumothorax, radiographer, artifact

## Abstract

**Objectives:** We aimed to investigate whether AI assistance could improve the performance of pneumothorax detection on chest radiographs (CXR) by readers with varying experience from radiologists to the frontline healthcare providers, and whether AI assistance could diminish the potential confounders for readers’ detecting pneumothorax. **Methods:** In this retrospective, single-center, blinded, multi-reader diagnostic accuracy study, 125 CXRs were prepared from radiological information system (March 2024 to August 2024) for test. The 18 readers were composed of six groups, each had 3 persons: board-certified radiologists (Group-1), senior radiology residents (Group-2), junior radiology residents (Group-3), postgraduate year residents (Group-4), senior radiographers (Group-5), and junior radiographers (Group-6). They read the CXR independently twice, without and with AI assistance, at an interval of one month. We used receiver operating characteristic curve for performance analysis and generalized estimating equation (GEE) model for confounding factor analysis. **Results:** AI software alone achieved a high area under curve of 0.965 (95% CI: 0.926, 0.995). With AI assistance, the performance in all groups significantly improved (*p* < 0.01) especially the junior readers (the frontline healthcare providers, Group-3, 4, 6) and diminished the difference among all groups except some related to Group-1. GEE model showed that AI assistance, reader’s experience, and projection type interfere with the readers’ performance (all *p* < 0.05). **Conclusions:** AI assistance could improve the performance of pneumothorax detection by varying experience of readers, especially the frontline healthcare providers. The influence of confounders, such as reader’s experience, also be diminished by AI assistance.

## 1. Introduction

Pneumothorax is one of the most common thoracic emergencies, which can be life-threatening and require prompt diagnosis for clinical management. Chest radiography (CXR) is regarded as the primary imaging modality for first-line assessment. However, interpreting CXRs depends heavily on the reader’s expertise and can be challenging due to subtle imaging findings or presence of artifacts [1,2].

In recent years, artificial intelligence (AI) systems have been introduced for image interpretation, showing an improvement in diagnostic accuracy [3,4,5,6]. Nevertheless, AI models also have their limitations. Common sources of false positives for detecting pneumothorax were described in previous studies [3,7,8], including rib edges, cardiac borders, pulmonary bullae, gastric gas, and linear artifacts such as thoracic tubes or skin folds.

Several studies have evaluated the performance of AI-assisted detection in pneumothorax for radiologists [9,10,11] and clinicians [12], but few have addressed its impact on radiographers and post-graduate year (PGY) residents, who are often at the frontline of assessment and are less experienced in interpreting emergency CXRs. Furthermore, the influence of confounding factors such as pneumothorax size and skinfold artifacts on AI-assisted interpretation remains insufficiently explored.

This study aimed to assess the diagnostic performance of an AI model for pneumothorax detection on chest radiographs across healthcare providers with varying levels of expertise. Additionally, we investigated the effects of pneumothorax severity and the presence of skinfold artifacts as potential confounding factors within this context.

## 2. Materials and Methods

The ethics committee of Kaohsiung Veterans General Hospital (approval number KSVGH-25-CT3-05) approved this study. The need for informed consent was waived. No authors had a conflict of interest to disclose.

### 2.1. Image Dataset

Inpatient CXRs from the radiology information system of Kaohsiung Veterans General Hospital from March 2024 to August 2024 were retrospectively chosen by a 25-year-experience thoracic radiologist to generate a dataset of 125 radiographs without or with different severity of pneumothorax. Artifacts such as skin folds were intentionally overrepresented to enrich the dataset and ensure adequate inclusion of diagnostically challenging cases. Enrichment strategy was used to retrospectively select CXR from inpatient department and portable CXR with key words of “pneumothorax” and “skin fold” form the RIS reports, respectively, to enroll sufficient and balanced cases number for each group. CXRs with uncertain medical records or lack of follow-up CXRs were excluded. The severity of pneumothorax was defined as small (<2 cm) and large (>2 cm) according to the British Thoracic Society (BTS) guidelines (2010) [13] (Figure 1).

### 2.2. AI Software

An FDA-approved AI assistance CXR software (Lunit Insight CXR ver 3.1.4.1; Lunit Inc., Seoul, Republic of Korea) was applied in this study. This software is designed to detect 10 common abnormalities (pneumothorax, mediastinal widening, pneumoperitoneum, nodule/mass, consolidation, pleural effusion, linear atelectasis, fibrosis, calcification and cardiomegaly) in CXRs. The AI software assists readers to interpret CXRs by marking the suspected abnormal areas, labeling the type of abnormality, and generating heat maps indicating the probability of each abnormality on a second capture of CXR, with likelihood percentages ranging from 1% to 100%. Its performance has been validated in previous studies [14,15].

### 2.3. Readers

Eighteen readers participated in this study. They were composed of six groups, each consisting of three participants: board-certified radiologists (Group 1), senior radiology residents (Group 2, in their 4th year of residency), junior radiology residents (Group 3, in their 1st to 2nd year of residency), postgraduate year (PGY) residents (Group 4), senior radiographers with more than 5 years of experience (Group 5), and junior radiographers with less than 1 year of experience (Group 6). For analysis of impact of years of experience, these 6 groups were further categorized into senior group (Group 1, 2, 5) and junior group (Group 3, 4, 6).

### 2.4. Scoring

The CXRs were presented in a randomly arranged order, and all readers were blinded to the clinical information. Each reader independently assessed the presence of pneumothorax using a 5-point Likert scale, rating the likelihood of pneumothorax from 0 to 4 (0 = definitely no, 1 = probably no, 2 = equivocal, 3 = probably yes, 4 = definitely yes), without AI assistance during the first reading session. After a one-month interval, the same process was repeated at another randomly arranged order with AI assistance, i.e., the CXR and heat map of abnormality were present simultaneously for interpretation.

### 2.5. Reference Standard

The reference standard was the judgment of a 25-year-experience board-certified thoracic radiologist, based on available serial CXR, CT scan and clinical chart records.

### 2.6. Statistical Analyses

We used R software (version 4.4.3) to perform all statistical analyses. Diagnostic accuracy for both AI model and readers was assessed using the Area Under the Receiver Operating Characteristic Curve (AUROC; AUC). Mean AUCs with and without AI assistance were compared using two-sided 95% confidence intervals (CIs). Pairwise AUC comparisons were performed using DeLong’s test with Holm’s correction for multiple comparisons, with statistical significance defined as *p* < 0.05. The secondary outcome was to analyze the probable influence of those potential confounding factors. A generalized estimating equations (GEE) model was used to account for the within-subject correlation arising from multiple diagnostic assessments performed on the same patient (i.e., repeated observations by different readers). The results were demonstrated as parameter estimates along with their corresponding standard errors (SE) and *p*-value. A *p*-value of less than 0.05 indicated a statistical significance. Three primary variables were included in the model: (1) the presence or absence of AI assistance, (2) reader experience level (senior vs. junior), and (3) the presence of skinfold artifacts (coded as 0 for absent and 1 for present). Two additional variables were entered as covariates to control for potential confounding: pneumothorax volume (coded as 0 for no pneumothorax, 1 for small, and 2 for large pneumothorax) and projection type (coded as 0 for PA, 1 for AP, and 2 for portable views). All main effects and interaction terms among the primary variables were included in the analysis.

## 3. Results

### 3.1. Data Characteristics

A total of 125 CXRs from 125 patients were selected. The demography and CXR characteristics details are listed in Table 1. There were 88 male and 37 female patients, with a mean age of 66.4 years. Among these, 73 (58.4%) had no pneumothorax and 52 (41.6%) had pneumothorax (30 small, 22 large). 61 were obtained in supine AP projection, 16 in standing AP and 48 were in standing PA views. Skinfold artifacts were present in 41 CXRs (Table 1).

### 3.2. Performance of AI Alone and Compared with Unassisted Readers

The AUC of AI alone was 0.965 (95% CI: 0.926, 0.995), which was significantly higher than (*p* < 0.05) all the readers’ group except for Group 1 (Figure 2; Table 2a,b).

### 3.3. Readers’ Performance Without AI Assistance

The AUC of each group without AI assistance was Group 1, 0.908 (95% CI: 0.874, 0.935); Group 2, 0.850 (95% CI: 0.810, 0.885); Group 3, 0.777 (95% CI: 0.731, 0.818); Group 4, 0.711 (95% CI: 0.662, 0.757); Group 5, 0.840 (95% CI: 0.798, 0.875); Group 6, 0.607 (95% CI: 0.556, 0.657) (Figure 3a–f). Significant differences were observed between Group 1 and Groups 3, 4, and 6; between Group 2 and Groups 4 and 6; between Group 3 and Group 6; between Group 4 and Group 6; and between Group 5 and Group 6 (*p* < 0.01) (Table 2a).

### 3.4. Readers’ Performance with AI Assistance

With AI assistance, the performance of all groups improved (Figure 3a–f and Figure 4a,b). The AUC of each group with AI assistance was of Group 1, 0.933 (95% CI, 0.902 to 0.95); Group 2, 0.889 (95% CI, 0.853 to 0.919); Group 3, 0.864 (95% CI, 0.825 to 0.818); Group 4, 0.832 (95% CI, 0.79 to 0.868); Group 5, 0.879 (95% CI, 0.841, 0.910); Group 6, 0.837 (95% CI, 0.796 to 0.873). No significant differences were found between Group 1 and Groups 2, 3, and 5, and no significant pairwise differences were observed among Groups 2 to 6 after AI assistance (Table 2b).

Take two supine chest radiographs for examples (Figure 5). Figure 5a is a case of pneumothorax presents with the deep sulcus sign. Without AI assistance, apart from the board-certified radiologist, all other groups were not definite certain about the presence of pneumothorax, and junior radiographers were even unable to detect it. However, with AI assistance, all readers were able to confidently identify pneumothorax, reaching a median scale of 4 in every subgroup. Figure 5b is a case of supine radiograph with skinfold artifacts which mimic pneumothorax for junior groups. Even with AI assistance, some junior radiographers still insisted on the presence of pneumothorax. These two cases, respectively, show that AI assistance improved the false negative readings (Figure 5a) and the false positive readings (Figure 5b) in detecting pneumothorax, especially for the junior group (Group 4, Group 6).

### 3.5. GEE Model of Pneumothorax Diagnostic Accuracy of Readers

#### 3.5.1. Main Effects

Analysis of the main effects revealed several significant findings. First, AI assistance significantly improved diagnostic accuracy (estimate = 0.76, SE = 0.14, Wald = 31.2, *p* < 0.001). Second, reader experience was also a strong predictor, with senior readers outperforming junior readers (estimate = 0.94, SE = 0.13, Wald = 49.5, *p* < 0.001). Among covariates, diagnostic accuracy varied by pneumothorax severity relative to large pneumothorax. When no pneumothorax was present, accuracy was higher (ptx012 = 0: estimate = 1.56, SE = 0.13, Wald = 147.2, *p* < 0.001), whereas small pneumothorax was more difficult to identify correctly (ptx012 = 1: estimate = −1.16, SE = 0.11, Wald = 116.1, *p* < 0.001). For projection type of image (relative to the reference category PA), the AP view was not statistically significant (estimate = 0.11, SE = 0.16, *p* = 0.50), while portable view was associated with lower accuracy (estimate = −0.33, SE = 0.10, Wald = 9.9, *p* = 0.002). The presence of skinfold artifacts (SF) did not show a significant main effect in this model (sf = 1: estimate = −0.16, SE = 0.19, *p* = 0.40). See Table 3 for full results. (Table 3). To further examine the issue of skinfold artifacts (SF), we conducted a descriptive analysis of diagnostic outcome distributions stratified by SF status, Seniority, and AI assistance. Without AI assistance, the false positive rate (FPR) was 5% in the junior group and 1% in the senior group when SF is present, and 3% versus 1% without SF. The false negative rate (FNR) was 7% in the junior group and 5% in the senior group with SF, and 15% versus 9% without SF. With AI assistance, the FPR decreased to 2% in the junior group and 1% in the senior group with SF, and 2% versus 1% without SF. The FNR was 5% versus 4% with SF, and 10% versus 7% without SF. (Table 4).

#### 3.5.2. Two-Way Interactions

A significant two-way interaction was observed between AI assistance and reader experience (estimate = −0.43, SE = 0.20, Wald = 4.7, *p* = 0.03). This negative interaction indicates that the improvement associated with AI assistance was greater for junior readers than for senior readers (i.e., AI narrows the senior–junior performance gap). Interactions involving skinfold artifacts were not significant (AI assistance × Skinfold artifacts: *p* = 0.90; Senior group × Skinfold artifacts: *p* = 0.29). Predicted accuracies illustrating these patterns across AI aid and skinfold status by reader experience are shown in Figure 6, with 95% confidence intervals.

#### 3.5.3. Higher-Order Interactions

No higher-order interaction reached statistical significance. In particular, the three-way interaction among AI assistance, reader experience, and skinfold status was not significant (AI assistance × Senior group × Skinfold artifacts: *p* = 0.50), suggesting no additional modifying effect beyond the AI × experience interaction.

## 4. Discussion

Our results confirmed that the applied AI model is a reliable tool for detecting pneumothorax. With AI assistance, diagnostic variability among the six reader groups was markedly reduced, including, for the first time in the literature report, the radiographer and PGY residents. The improvement was particularly pronounced among junior readers, who are often at the first line in clinical workflow.

Previous studies have reported high diagnostic performance for AI algorithms for pneumothorax detection on CXR, with AUC values ranging from 0.88 to 0.98 [12,16,17,18,19,20,21]. A meta-analysis of 23 studies conducted in 2024 revealed relatively high sensitivity and specificity for pneumothorax detection using deep learning algorithms [21], with the pooled sensitivity and specificity being 87% (95% CI: 0.81, 0.92) and 95% (95% CI: 0.92, 0.97%), respectively. Our observed AUC of 0.965 (95% CI: 0.926, 0.995) for pneumothorax detection aligns with previously published validations of Lunit Insight CXR, such as Ahn et al. [22], who reported an AUC of 0.999 for AI stand-alone detection for pneumothorax, and van Beek et al. (2023) [23], who found an AUC of 0.954 in primary and emergency settings. These results indicate that the excellent diagnostic performance in our study is consistent with real-world validation studies.

The benefits of AI-assisted interpretation have been widely reported for radiologists [9,10,11] and clinicians [12], particularly for less experienced readers [11,12]. However, few studies have explored its impact on PGY residents and radiographers, who play a critical role as frontline providers.

In our results, readers’ diagnostic accuracy without AI assistance closely correlated with experience level. Notably, senior radiographers performed comparably to senior radiology residents, suggesting that accumulated clinical exposure may compensate for formal training differences. AI assistance effectively elevated junior radiographers’ performance to a level comparable with their senior counterparts. This improvement is clinically meaningful, as radiographers and PGY residents often serve as the earliest detectors of critical findings such as pneumothorax in urgent settings.

To enhance communication efficiency in emergency radiology, the early alert role of radiographers has been emphasized in several studies [24,25,26]. In our institution, radiographers are trained to assess image quality and to promptly recognize critical findings such as barotrauma or tube malposition. There was meta-analytic evidence supporting that radiographers can accurately interpret plain radiographs, with performance comparable to radiologists in accident and emergency settings [27,28]. However, most previous research focused on trauma or lung cancer screening, with limited data on pneumothorax detection in critical care and almost none on very junior radiographers, who are often responsible for portable CXRs. To our knowledge, this is the first study examining pneumothorax detection by junior radiographers and PGY residents, comparing them to senior readers while also evaluating the effect of AI assistance. AI support enabled junior radiographers to perform at a level comparable to senior groups, reinforcing their potential to act as first responders in identifying pneumothorax. Khalaf et al. (2025) [29] investigated the knowledge, expectations, and attitudes toward AI integration in medical imaging in Kuwait. While most respondents showed a positive outlook, the study identified a need for greater AI-oriented education and training. In line with our findings, AI could function not only as a diagnostic support tool but also as an educational resource. With proper integration into training curricula, AI systems may accelerate learning curves, improve diagnostic confidence, and empower radiographers and PGY residents to take more active roles in emergency and routine image interpretation.

According to our clinical experience, certain confounding factors might bother readers when interpreting CXRs for detecting pneumothorax. In fact, this retrospective study was inspired by the frequent consultation of CXRs that have skinfold artifacts, and therefore, we intentionally enrolled more such cases (56/125 cases). However, GEE analysis revealed that AI assistance, reader’s experience, pneumothorax amount, and portable view were the main effects, while the presence of skinfold artifacts did not seem to significantly influence the diagnostic accuracy. A descriptive analysis of diagnostic outcomes stratified by skinfold (SF) status, reader’s seniority, and AI assistance showed that without AI assistance, junior readers showed higher false positive rates (FPR) and false negative rates (FNR) than senior readers, regardless of the presence of skinfold artifacts. With AI support, both rates decreased, and the performance gap between groups narrowed, suggesting that AI had a particularly beneficial effect on less experienced readers. Interestingly, the presence of skinfold artifacts appeared to slightly reduce the FNR, possibly because the visible artifact prompted readers to more carefully assess for pneumothorax, thereby reducing false negatives in both groups. However, skinfold artifacts were not a statistically significant factor in the revised GEE model. This aligns with prior evidence suggesting that well-trained AI models are relatively resistant to common image artifacts. For instance, Hillis et al. (2022) [17] reported consistently high AUCs for pneumothorax detection in both cases with (AUC = 0.964) and without (AUC = 0.982) skinfold artifacts, indicating robust algorithmic performance. These results support the robustness of AI models to common imaging confounders and emphasize their potential as both supportive and educational tools, particularly in resource-limited or high-acuity environments where less experienced readers often interpret chest radiographs.

For interaction analysis, the results revealed a significant two-way interaction between AI assistance and reader experience, indicating that AI provided a greater diagnostic benefit for junior readers compared with senior readers. This finding suggests that AI support can effectively narrow the performance gap between less experienced and more experienced readers, particularly in detecting subtle or small pneumothoraxes. Such results align with previous research showing that AI-assisted image interpretation improves diagnostic accuracy mainly among less experienced clinicians, while providing limited additional benefit to experts who already perform at a high baseline level [12,22].

The study has several limitations. First, the dataset was intentionally enriched with challenging cases, meaning results may differ from those in a true screening population. Nevertheless, since this study was inspired by frequent clinical misinterpretations involving skinfold artifacts, a balanced case distribution was necessary to assess the statistical impact of these artifacts. Second, the study’s relatively small sample size and retrospective single-center design may limit generalizability. Third, all readers were from the same institution, which introduces the possibility of institutional bias. Future multicenter and prospective studies are warranted to validate these findings. Nonetheless, our inclusion of 18 readers across multiple professional levels provides valuable preliminary evidence to inform future large-scale designs and potential real-world implementation of AI-assisted interpretation within PACS systems.

## 5. Conclusions

We confirmed that AI-assisted reading improves pneumothorax detection accuracy on CXR across diverse experience levels and remains robust against confounding artifacts. This study extends AI validation to the frontline of clinical workflow, encompassing radiographers and PGY residents for the first time. We believe these extensions closely reflect the impact of AI-assisted CXR reading in real-world scenarios nowadays, and our findings suggest that AI not only enhances diagnostic accuracy but may also serve as a training and decision-support tool to strengthen emergency radiological care. Future prospective randomized controlled trials are recommended to assess the cost-effectiveness and scalability of integrating AI assistance into real-world frontline practice.

## Figures and Tables

**Figure 1 diagnostics-15-02639-f001:**
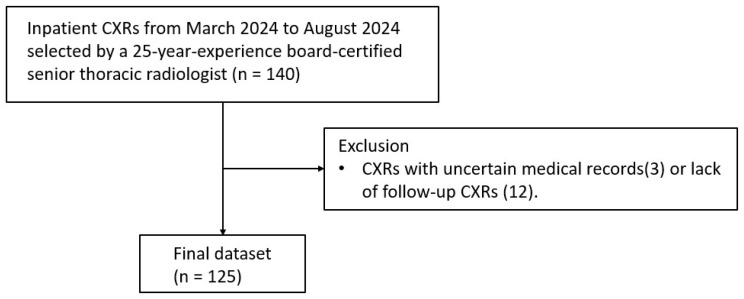
Study flowchart.

**Figure 2 diagnostics-15-02639-f002:**
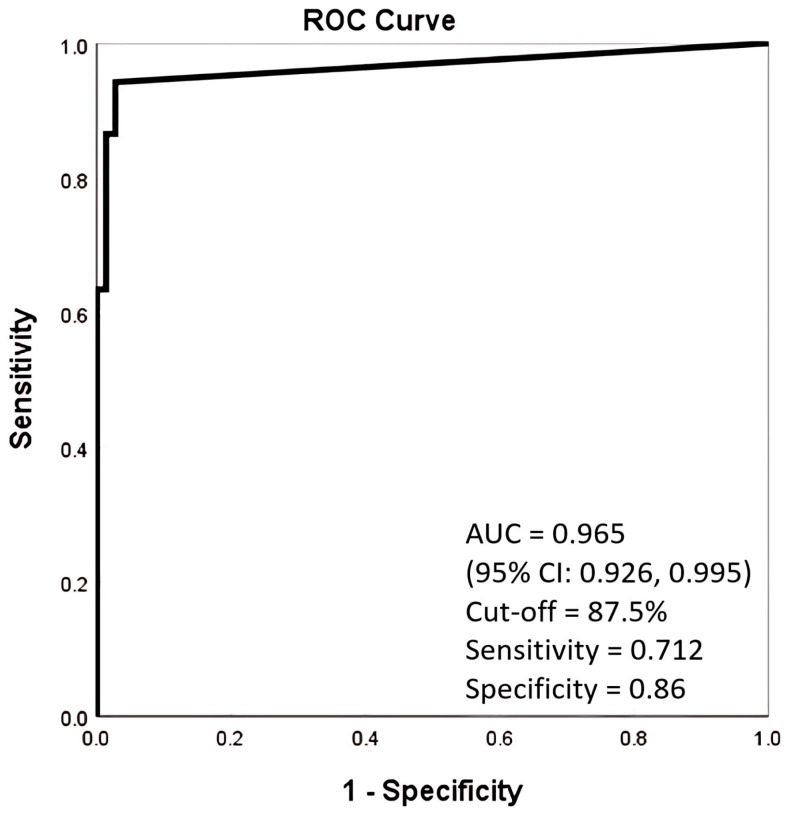
ROC curve of AI detecting pneumothorax.

**Figure 3 diagnostics-15-02639-f003:**
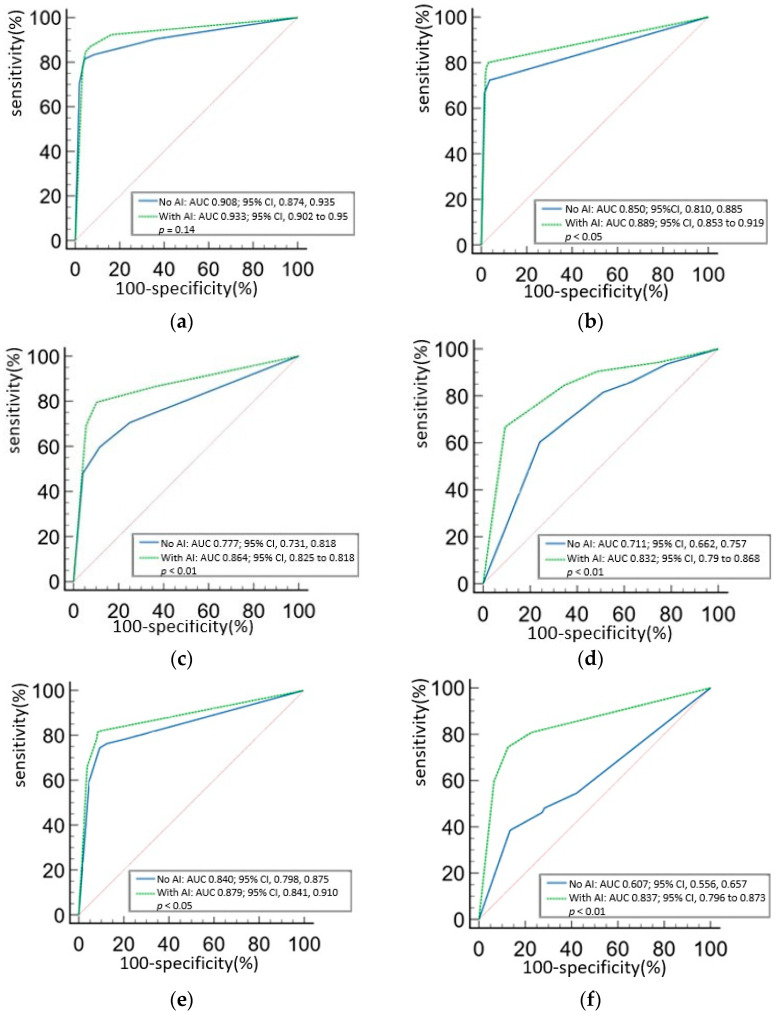
ROC curve of each group, with and without AI-assistance (**a**) Group 1, (**b**) Group 2, (**c**) Group 3, (**d**) Group 4, (**e**) Group 5, (**f**) Group 6. Group 1, board-certified radiologists; 2, senior radiology residents; 3, junior radiology residents; 4, postgraduate year residents; 5, senior radiographers; 6, junior radiographers.

**Figure 4 diagnostics-15-02639-f004:**
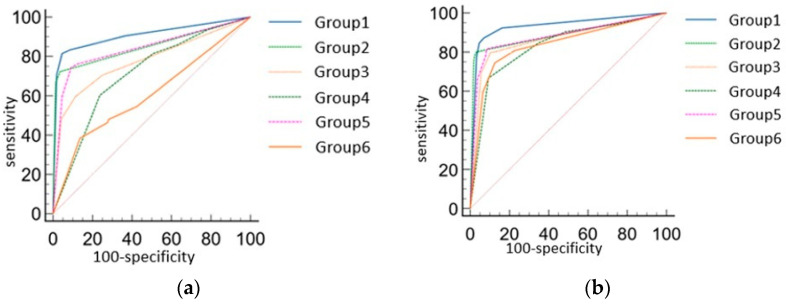
(**a**) The comparison of ROC curve of AI and each group without AI assistance; (**b**) The comparison of ROC curve of each group with AI assistance.

**Figure 5 diagnostics-15-02639-f005:**
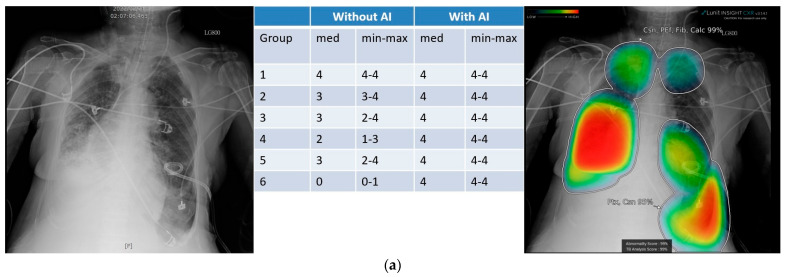
Two examples to illustrate the impact of AI assistance on chest radiograph reading. (**a**) Supine chest radiograph of a 73-year-old male with massive pneumothorax (deep sulcus sign); (**b**) Supine chest radiograph of a 77-year-old man with skin fold artifact. Left panel, the native radiograph; right panel, with AI-heat maps and labels; middle panel: the score sheet without and with AI of each group. Group 1, board-certified radiologists; 2, senior radiology residents; 3, junior radiology residents; 4, postgraduate year residents; 5, senior radiographers; 6, junior radiographers. med: median; min–max: minimal to maximal.

**Figure 6 diagnostics-15-02639-f006:**
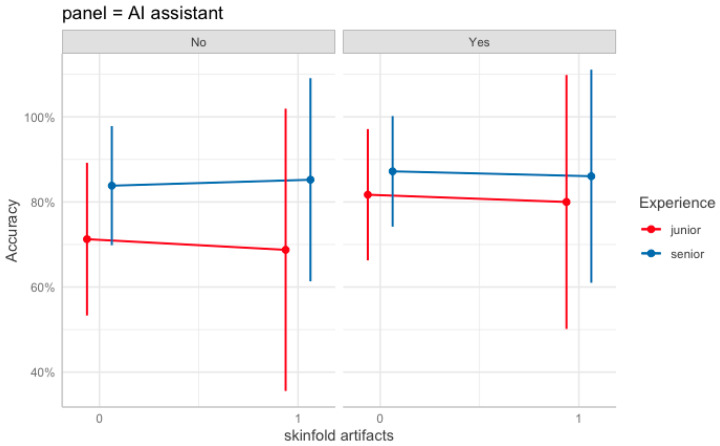
Panels represent the presence (“Yes”) or absence (“No”) of AI assistance. The *x*-axis denotes the presence of skinfold artifacts (0 = absent, 1 = present), and lines depict diagnostic accuracy for junior (red) and senior (blue) readers. Vertical bars represent 95% confidence intervals of the predicted values. The model controlled for pneumothorax severity (ptx012) and projection type.

**Table 1 diagnostics-15-02639-t001:** Patient demographics.

Inpatient CXRs (*n* = 125, 125 CXRs from 125 Patients)			
Mean age ± standard deviation	66.36 ± 19.2		
Gender			
	Male	88 (70.4%)	
	Female	37 (29.6%)	
Pneumothorax			
	Absent	73 (58.4%)	
	Present	52 (41.6%)	
		Small (<2 cm)	30 (57.7%)
		Large (≥2 cm)	22 (42.3%)
Projection type			
	Upright PA	48 (38.4%)	
	Upright AP	16 (12.8%)	
	Supine AP	61 (48.8%)	
Skinfold artifacts			
	Present	56 (44.8%)	
		No pneumothorax	38 (67.9%)
		Small pneumothorax	12 (21.4%)
		Large pneumothorax	6 (10.7%)
	Absent	69 (55.2%)	

**Table 2 diagnostics-15-02639-t002:** The comparison of AUC of each group and *p* value of difference (with Holm’s correction) between AI and each group, (**a**) without and (**b**) with AI assistance.

**(a)**							
	AI	Group 1	Group 2	Group 3	Group 4	Group 5	Group 6
AUC	0.965	0.907	0.85	0.776	0.713	0.84	0.606
*p* value							
Group 1	0.14						
Group 2	<0.01	0.14					
Group 3	<0.01	<0.01	0.11				
Group 4	<0.01	<0.01	<0.01	0.15			
Group 5	<0.01	0.09	0.71	0.15	<0.01		
Group 6	<0.01	<0.01	<0.01	<0.01	0.04	<0.01	
**(b)**							
		Group 1	Group 2	Group 3	Group 4	Group 5	Group 6
AUC		0.933	0.889	0.864	0.832	0.879	0.837
*p* value							
Group 1							
Group 2		0.49					
Group 3		0.07	>0.99				
Group 4		<0.01	0.40	>0.99			
Group 5		0.22	>0.99	>0.99	0.76		
Group 6		<0.01	0.49	>0.99	>0.99	0.94	

**Table 3 diagnostics-15-02639-t003:** GEE analysis of confounding factors.

	Estimate	SE	Wald	Sig.
(Intercept)	0.65	0.12	30.0	***
AI assistance (Yes)	0.76	0.14	31.2	***
Senior group	0.94	0.13	49.5	***
No pneumothorax	1.56	0.13	147.2	***
Small pneumothorax	−1.16	0.11	116.1	***
Skinfold artifacts	−0.16	0.19	0.7	
Projection type: AP	0.11	0.16	0.4	
Projection type: portable	−0.33	0.10	9.9	**
AI assistance × Senior group	−0.43	0.2	4.7	*
AI assistance × SF	0.02	0.27	0.0	
Senior group × SF	0.29	0.27	1.2	
AI assistance × Senior group × Skinfold artifacts	−0.27	0.40	0.4	

Note. ‘:’ indicates the interaction between variables; * *p* < 0.05, ** *p* < 0.01, *** *p* < 0.001.

**Table 4 diagnostics-15-02639-t004:** Descriptive analysis of diagnostic outcome distributions stratified by skinfold artifacts, reader’s seniority and AI assistance.

Seniority	AI Aid	Diagnosis Outcome	SF (−)	SF (+)
Senior	Yes	True negative	27%	30%
		False negative	7%	4%
		True positive	20%	10%
		False positive	1%	1%
	No	True negative	27%	30%
		False negative	9%	5%
		True positive	18%	9%
		False positive	1%	1%
Junior	Yes	True negative	26%	28%
		False negative	10%	5%
		True positive	18%	10%
		False positive	2%	2%
	No	True negative	25%	26%
		False negative	15%	7%
		True positive	13%	8%
		False positive	3%	5%

## Data Availability

The data presented in this study are available on request from the corresponding author; the data are owned by the hospital and cannot be publicly released without formal permission.

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
