# Peer review of "Impact of AI Assistance in Pneumothorax Detection on Chest Radiographs Among Readers of Varying Experience"

_diagnostics, 2025, doi:10.3390/diagnostics15202639_

Round 1

Reviewer 1 Report

Comments and Suggestions for Authors

Methodology and Study Design

1. Could the authors provide more detailed justification for the specific selection criteria used by the 25-year-experience thoracic radiologist when choosing the 125 CXRs, particularly regarding how representativeness of real-world cases was ensured?

2. How did the authors ensure that the one-month washout period between reading sessions was adequate to prevent recall bias, and were there any measures taken to verify that readers did not remember specific cases?

3. Could the authors elaborate on why skin fold artifacts were intentionally over-represented (56/125 cases, 44.8%) in the dataset and how this proportion compares to typical clinical practice prevalence?

Statistical Analysis and Results

4. The authors mention using DeLong's test for AUC comparisons but do not provide correction for multiple comparisons; could they clarify whether any adjustment was applied and justify this methodological choice?

5. In the GEE model, could the authors explain why higher-order interactions (three-way and four-way) were included in the analysis despite none reaching statistical significance, and whether model parsimony was considered?

6. The paper reports that skinfold artifacts were "positively associated with correct diagnosis" (estimate = 1.037), which seems counterintuitive; could the authors provide a more detailed explanation of this finding?

Study Population and Generalizability

7. Could the authors discuss whether the relatively small sample size of three readers per group (n=3) provides sufficient statistical power to detect meaningful differences between groups, particularly for the more experienced readers?

8. How do the authors account for potential institutional bias, given that all readers were from a single center, and what implications might this have for generalizability to other healthcare settings?

9. The paper includes both supine and upright chest radiographs in various projections; could the authors clarify whether projection type was controlled for in the analysis or considered as a potential confounding factor?

AI System Performance and Validation

10. Could the authors provide more information about the training dataset and validation methodology of the Lunit Insight CXR software, particularly whether it was trained on cases similar to those used in this study?

11. The AI achieved an exceptionally high AUC of 0.965; could the authors discuss whether this performance level is consistent with the software's performance in other published studies and real-world implementations?

Clinical Implications and Implementation

12. Given that junior radiographers showed substantial improvement with AI assistance, could the authors elaborate on the practical workflow implications and potential liability considerations for implementing such a system in clinical practice?

13. The authors suggest extending AI assistance to frontline healthcare providers; could they provide more specific recommendations about training requirements and quality assurance measures that would be necessary for such implementation?

Data Interpretation and Limitations

14. The study found no significant three-way or four-way interactions involving AI assistance; could the authors discuss whether this suggests that AI assistance provides uniform benefit regardless of reader experience and case complexity, or whether the study was underpowered to detect such interactions?

15. Could the authors expand on how the retrospective design and intentional enrichment of challenging cases (skin fold artifacts, varying pneumothorax sizes) might affect the external validity of their findings in routine clinical practice?

Comments on the Quality of English Language

The English could be improved to more clearly express the research.

Author Response

Response to Reviewer 1 Comments

1. Summary

We sincerely thank Reviewer 1 and editor for their thorough review and constructive feedback. Their insightful comments have helped us identify several aspects that required clarification and refinement, leading to substantial improvements in both the methodological transparency and overall scientific rigor of our manuscript. We are grateful for the reviewer’s detailed observations, which prompted us to re-examine and update our statistical analyses, clarify our sampling strategy, and enhance the discussion of AI-related implications. These revisions have made the manuscript more complete, coherent, and credible.

We have carefully addressed all comments point-by-point in the following section, with corresponding revisions clearly highlighted in the revised manuscript. We have carefully addressed all points raised, including clarifications on case selection criteria, justification of the washout period, explanation of purposive sampling with skinfold enrichment, updates to the statistical analyses (including DeLong’s test with Holm’s correction and a revised GEE model), additional discussion of AI performance and workflow implications, and acknowledgment of study limitations regarding sample size and institutional bias..

2. Questions for General Evaluation

Reviewer’s Evaluation

Response and Revisions

Does the introduction provide sufficient background and include all relevant references?

Can be improved

We reviewed our references to ensure their relevance. Additional literature supporting our study was identified, and we feel it fits better in the Discussion section.

Is the research design appropriate?

Can be improved

The study design was clarified with added justification for the one-month washout period and the inclusion of skinfold-enriched cases in point-by-point response.

Are the methods adequately described?

Must be improved

The methods section has been revised for greater clarity, including detailed descriptions of the cases selection and GEE model setup. (See response 1 to response 5.)

Are the results clearly presented?

Can be improved

We have reanalyzed the data with updated statistical methods and presented the results in a more concise and structured manner. The revised Results section now focuses on key findings with clearer explanation of significant outcomes. (See response 6, 7, and 9.)

Are the conclusions supported by the results?

Can be improved

According to the new results, we revise our discussion and conclusion, and some figures and tables have been reformatted for clarity.

Are all figures and tables clear and well-presented?

Can be improved

Several figures and tables that were previously unclear have been revised for improved clarity, and additional tables have been incorporated to more effectively illustrate and support our conclusions.

3. Point-by-point response to Comments and Suggestions for Authors

Comments 1: Could the authors provide more detailed justification for the specific selection criteria used by the 25-year-experience thoracic radiologist when choosing the 125 CXRs, particularly regarding how representativeness of real-world cases was ensured?

Response 1: Enrichment strategy (See response 3) was used to retrospectively select CXR from inpatient department and portable CXR with key words of “pneumothorax” and “skin fold” form the RIS reports, respectively, to enroll sufficient and balanced cases number for each group. The CXR reports were verified by the radiologist using a serial CXR findings or CT scan findings and clinical chart of each case as standard reference. We also added this information in our revised manuscript (page 2, Materials and Methods section, line 69-74).

Comments 2: How did the authors ensure that the one-month washout period between reading sessions was adequate to prevent recall bias, and were there any measures taken to verify that readers did not remember specific cases?

Response 2: We appreciate the reviewer’s insightful comment regarding the adequacy of the one-month washout period to minimize recall bias. When we designed our study, we referred to several prior works with similar multi-reader, multi-case designs. For example, Jin et al. (2022) adopted a four-week washout period and Ahn et al. (2022) also adopted a four-week washout period. After reviewing their methodologies, we applied a comparable four-week (one-month) washout in our study.

Although we did not systematically check whether readers remembered specific cases after the washout period, we did do some sample cases testing before the formal second-run reading. No report of residual memory of specific cases was recorded. Therefore, we thought it would be feasible to do the protocol of 4-week wash out.

Thank you for this valuable suggestion, and we agree that incorporating a formal verification measure would strengthen future studies.

References:

1.     Jin, K.N. et al. Diagnostic effect of artificial intelligence solution for referable thoracic abnormalities on chest radiography: a multicenter respiratory outpatient diagnostic cohort study. Eur Radiol 2022, 32, 3469–3479. doi:10.1007/s00330-021-08397-5.

2.     Ahn, J.S.; Ebrahimian, S.; McDermott, S.; Lee, S.; Naccarato, L.; Di Capua, J.F.; Wu, M.Y.; Zhang, E.W.; Muse, V.; Miller, B.; et al. Association of Artificial Intelligence-Aided Chest Radiograph Interpretation With Reader Performance and Efficiency. JAMA Netw Open 2022, 5, e2229289, doi:10.1001/jamanetworkopen.2022.29289.

Comments 3: Could the authors elaborate on why skin fold artifacts were intentionally over-represented (56/125 cases, 44.8%) in the dataset and how this proportion compares to typical clinical practice prevalence?

Response 3: Thank you for this insightful question. The potential influence of skinfold artifacts on AI-based pneumothorax detection was a key motivation for our study. We adopted a purposive sampling strategy to ensure adequate representation of challenging cases, particularly chest radiographs containing skinfold artifacts. As this was a relatively small, single-center study, purposive enrichment allowed us to meaningfully assess the diagnostic impact of these confounding factors on both AI and human readers. Although the proportion of skinfold artifacts (44.8%) was higher than in routine clinical practice, this was intentional and consistent with established practices in diagnostic accuracy and AI validation studies. Such enrichment of low-prevalence or confounding cases helps achieve sufficient statistical power for subgroup and interaction analyses, while also stress-testing both readers and algorithms under diagnostically uncertain conditions.

Enrichment strategies have been widely adopted in diagnostic studies. For instance, Homayounieh et al. (2021) intentionally included pulmonary nodules of varying difficulty to assess AI and reader performance across different detection challenges, and Jones et al. (2022) used an enriched dataset to evaluate the effectiveness of abbreviated breast MRI. We have added corresponding clarifications to the Methods and Discussion (about limitation) sections of the revised manuscript (page 2, Materials and Methods section, line 69-74; and page 12, Discussion section, line 327-331).

References:

1.     Homayounieh, F.; Digumarthy, S.; Ebrahimian, S.; Rueckel, J.; Hoppe, B.F.; Sabel, B.O.; Conjeti, S.; Ridder, K.; Sistermanns, M.; Wang, L.; et al. An Artificial Intelligence-Based Chest X-ray Model on Human Nodule Detection Accuracy From a Multicenter Study. JAMA Netw Open 2021, 4, e2141096, doi:10.1001/jamanetworkopen.2021.41096.

2.     Jones, L.I.; Marshall, A.; Elangovan, P.; Geach, R.; McKeown-Keegan, S.; Vinnicombe, S.; Harding, S.A.; Taylor-Phillips, S.; Halling-Brown, M.; Foy, C.; et al. Evaluating the effectiveness of abbreviated breast MRI (abMRI) interpretation training for mammogram readers: a multi-centre study assessing diagnostic performance, using an enriched dataset. Breast Cancer Res 2022, 24, 55, doi:10.1186/s13058-022-01549-5.

Comments 4: The authors mention using DeLong's test for AUC comparisons but do not provide correction for multiple comparisons; could they clarify whether any adjustment was applied and justify this methodological choice?

Response 4: We re-analyzed the AUC comparisons using DeLong’s test with Holm’s correction for multiple comparisons (page 3, Materials and Methods section, line 112-114). The adjusted values showed slight numerical differences from the original results (page 5-6, Results section, table 2a and table 2b; significant changes are highlighted in red). The updated findings further reinforce our conclusion that AI assistance reduces diagnostic disparities between reader groups.

Comments 5: In the GEE model, could the authors explain why higher-order interactions (three-way and four-way) were included in the analysis despite none reaching statistical significance, and whether model parsimony was considered?

Response 5: Following the reviewer’s suggestion, we revised the model to adopt a more parsimonious specification. In the final model, we retained AI assistance, Reader’s experience, and skinfold artifacts as the primary variables of interest, along with their interaction term. In addition, we included two covariates (pneumothorax amount, and projection type) as control variables, without introducing interaction terms with these covariates (page 3-4, Materials and Methods section, line 120-127).

Comments 6: The paper reports that skinfold artifacts were "positively associated with correct diagnosis" (estimate = 1.037), which seems counterintuitive; could the authors provide a more detailed explanation of this finding?

Response 6: In the revised analysis, skinfold artifact (SF) was no longer a significant predictor in the GEE model. Although it reached statistical significance in the earlier analysis, its effect was unstable across model specifications, suggesting that the previous finding may not reflect a robust effect.

To further examine this issue, we conducted a descriptive analysis of diagnostic outcome distributions stratified by SF status, Seniority, and AI assistance (see table below). The FP was 5% with SF in junior group, higher than 1% in senior group, and it was also higher than 3% without SF in junior group. With AI assistance, both with or without SF decreased to 2% in junior group. These trends were compatible with our assumption, i.e. FP appears higher in junior group than in senior group, and improved by AI assistance in junior group only; However, they were not statistically significant in the revised GEE model.

Although our previous model reached statistical significance, its effect was weak (Est =1.037, SE = 0.359, p = 0.04) and unstable across model specifications, suggesting that the previous finding may not reflect a robust effect. We have added the above results, new table and the discussion to the results and discussion sections of the revised manuscript (page 8-10, Results section, line 205-245, table 3 and 4; page 11, Discussion section, line 299-326).

Seniority

AI aid

Diagnosis outcome

SF (-)

SF (+)

Senior

Yes

true negative

27%

30%

false negative

7%

4%

true positive

20%

10%

false positive

1%

1%

No

true negative

27%

30%

false negative

9%

5%

true positive

18%

9%

false positive

1%

1%

Junior

Yes

true negative

26%

28%

false negative

10%

5%

true positive

18%

10%

false positive

2%

2%

No

true negative

25%

26%

false negative

15%

7%

true positive

13%

8%

false positive

3%

5%

Comments 7: Could the authors discuss whether the relatively small sample size of three readers per group (n=3) provides sufficient statistical power to detect meaningful differences between groups, particularly for the more experienced readers?

Response 7: We would like to acknowledge that, due to the limitations of our hospital’s size, having only three readers per subgroup is indeed relatively small, which constitutes one of the limitations of our study. To address this, in further GEE analysis, we grouped the 18 readers into two broader groups, junior and senior, with 9 readers in each group, in order to achieve more meaningful results. While we acknowledge that larger samples would further enhance statistical power, our sample size was constrained by practical considerations of available personnel. Importantly, the number of clusters in our study is close to the commonly recommended threshold of 20 clusters for GEE analyses (Li et al., 2019), which indicates that the design might be sufficient to detect moderate to large effects, even if more subtle effects may not be fully captured.

Reference: Li, F., Forbes, A. B., Turner, E. L., & Preisser, J. S. (2019). Power and sample size requirements for GEE analyses of cluster randomized crossover trials. Statistics in medicine, 38(4), 636–649. https://doi.org/10.1002/sim.7995

Comments 8: How do the authors account for potential institutional bias, given that all readers were from a single center, and what implications might this have for generalizability to other healthcare settings?

Response 8: We acknowledge that because all readers were recruited from a single institution, it may introduce institutional bias and limit the generalizability of our findings. Our purpose is to evaluate the effect of readers’ experience, which we assume to be universally similar across the institutes. However, external validation in multicenter settings and across different healthcare systems will be essential to confirm the broader applicability of our results. We have added this point to the limitations section of the main text to inform readers of the potential institutional bias (page 12, Discussion section, line 332-334).

Comments 9: The paper includes both supine and upright chest radiographs in various projections; could the authors clarify whether projection type was controlled for in the analysis or considered as a potential confounding factor?

Response 9: Thank you for the important reminder. We have re-run the GEE analysis and included projection type as a control variable (page 3-4, Materials and Methods section, line 120-127). The new results and discussion of our analysis were added in our revised manuscript (page 8-10, Results section, line 205-245, table 3; page 11, Discussion section, line 299-302).

Comments 10: Could the authors provide more information about the training dataset and validation methodology of the Lunit Insight CXR software, particularly whether it was trained on cases similar to those used in this study?

Response 10: According to the official Lunit white paper, the training dataset of Lunit Insight CXR comprised a large, multi-institutional collection of chest radiographs labeled for a variety of thoracic abnormalities, including consolidation, nodules, cardiomegaly, pleural effusion, fibrosis, calcification, atelectasis, mediastinal widening, and pneumothorax.

The dataset contained tens of thousands of images per abnormality class to allow robust learning across varied presentations. In the development and validation paper by Nam et al. (2021), the authors described the use of a held-out validation cohort with rigorous ground truth labeling to confirm generalizability across institutions.

While the publicly disclosed training data do not specifically report the proportion of skinfold artifacts or the proportion of pneumothorax cases in training, the broad variety of abnormalities and the multi-site nature suggest that the model has seen a wide spectrum of imaging features. As such, we believe the AI model likely encountered cases with imaging artifacts during training, though not necessarily in the same enriched proportion as in our study.

References:

1.     Official Lunit white paper

2.     Nam, J.G.; Kim, M.; Park, J.; Hwang, E.J.; Lee, J.H.; Hong, J.H.; Goo, J.M.; Park, C.M. Development and validation of a deep learning algorithm detecting 10 common abnormalities on chest radiographs. Eur Respir J 2021, 57, doi:10.1183/13993003.03061-2020.

Comments 11: The AI achieved an exceptionally high AUC of 0.965; could the authors discuss whether this performance level is consistent with the software's performance in other published studies and real-world implementations?

Response 11: Our observed AUC of 0.965 for pneumothorax detection is in line with results reported in other external validations. For example, Ahn et al. (2022) demonstrated an AUC of 0.999 for AI stand-alone detection for pneumothorax, while van Beek et al. (2023) reported an AUC of 0.954 in a validation study conducted in primary and emergency care settings. These findings confirm that the high performance observed in our study is consistent with previously published evidence, although real-world results may vary depending on case mix and prevalence. We also added this information to the discussion paragraph (page 11, Discussion section, line 259-264).

Reference:

1.     Ahn, J.S.; Ebrahimian, S.; McDermott, S.; Lee, S.; Naccarato, L.; Di Capua, J.F.; Wu, M.Y.; Zhang, E.W.; Muse, V.; Miller, B.; et al. Association of Artificial Intelligence-Aided Chest Radiograph Interpretation With Reader Performance and Efficiency. JAMA Netw Open 2022, 5, e2229289, doi:10.1001/jamanetworkopen.2022.29289.

2.     van Beek, E.J.R.; Ahn, J.S.; Kim, M.J.; Murchison, J.T. Validation study of machine-learning chest radiograph software in primary and emergency medicine. Clin Radiol 2023, 78, 1-7, doi:10.1016/j.crad.2022.08.129.

Comments 12: Given that junior radiographers showed substantial improvement with AI assistance, could the authors elaborate on the practical workflow implications and potential liability considerations for implementing such a system in clinical practice?

Response 12: Thank you for raising this important point regarding the workflow and liability implications of implementing AI assistance, particularly for junior radiographers.

From a workflow perspective, our findings suggest that AI could play a supportive role in enhancing the diagnostic performance of frontline staff, including radiographers. This may allow earlier flagging of potential pneumothorax cases, thereby facilitating more timely review by radiologists and potentially reducing the diagnostic time in emergency settings. Similar considerations have been reported in prior work showing that trained radiographers can achieve diagnostic accuracy comparable to radiologists in plain radiograph reporting (Brealey et al., 2005), supporting the idea that empowering radiographers with AI tools may contribute to overall service efficiency.

With respect to liability, we agree that this is a crucial consideration. At present, AI systems are intended as decision-support tools rather than autonomous diagnostic agents. Thus, the ultimate responsibility for final interpretation should remain with licensed physicians or radiologists. We view AI assistance for junior radiographers as a mechanism to strengthen their preliminary evaluations and to reduce the time interval between examination and clinical management, rather than to replace the established reporting pathway. In future clinical deployment, clear institutional protocols, medico-legal frameworks, and continuous training will be necessary to ensure safe and responsible use.

Reference: Brealey, S.; Scally, A.; Hahn, S.; Thomas, N.; Godfrey, C.; Coomarasamy, A. Accuracy of radiographer plain radiograph reporting in clinical practice: a meta-analysis. Clin Radiol 2005, 60, 232-241, doi:10.1016/j.crad.2004.07.012.

Comments 13: The authors suggest extending AI assistance to frontline healthcare providers; could they provide more specific recommendations about training requirements and quality assurance measures that would be necessary for such implementation?

Response 13: Thank you for this important point. We are very encouraged by this study result. We have now added one-hour lecture of focused on the basic reading of emergency and portable CXR with the use of AI assistance on the e-learning system of our institute. The junior radiographer, PGY and junior radiological residence are required to learn the course and pass the examination on their e-portfolio. Monthly QC meeting of radiographer will include the cases of emergency CXR and portable CXR if feasible cases are available. The radiographers, no matter junior or senior, who made a correct early alarm of presence of pneumothorax or other critical findings of CXR will be scored and rewarded in our program.

Comments 14: The study found no significant three-way or four-way interactions involving AI assistance; could the authors discuss whether this suggests that AI assistance provides uniform benefit regardless of reader experience and case complexity, or whether the study was underpowered to detect such interactions?

Response 14: In our revised analysis (page 3-4, Materials and Methods section, line 120-127; page 8-10, Results section, line 205-245, table 3), we adopted a more parsimonious model that focuses on the three primary variables of interest (AI assistance, Reader’s experience and Skinfolds artifact) and their interaction, while adjusting for pneumothorax amount and projection type as covariates. This specification was chosen to enhance interpretability and to avoid overfitting, particularly given the limited sample size. Consequently, higher-order interactions (three-way or four-way) were not retained in the final model. In this final model, AI assistance improves diagnostic performance overall, and critically, the AI assistance × Seniority interaction indicates that the magnitude of benefit is larger for junior readers than for senior readers.

Comments 15: Could the authors expand on how the retrospective design and intentional enrichment of challenging cases (skin fold artifacts, varying pneumothorax sizes) might affect the external validity of their findings in routine clinical practice

Response 15: We acknowledge that the retrospective design and intentional enrichment of challenging cases may limit the direct generalizability of our findings to routine clinical practice where such cases are less frequent. However, our aim was to stress-test both human readers and the AI algorithm under conditions that are particularly prone to diagnostic error. This approach is consistent with prior validation strategies for diagnostic algorithms (as we mentioned on comment 4), in which enrichment of difficult cases is used to permit meaningful subgroup and interaction analyses.

While this design may not fully reflect real-world prevalence, it allows us to highlight situations where AI assistance is most valuable and where readers are most likely to struggle. Larger multicenter, prospective studies with representative case distributions will be required to further confirm the external validity of our findings.

4. Response to Comments on the Quality of English Language

Point 1: The English could be improved to more clearly express the research.

Response 1: Thank you for your opinion. We have carefully re-reviewed the entire manuscript and made English language revisions to improve readability and clarity for the readers.

5. Additional clarifications

NA

Reviewer 2 Report

Comments and Suggestions for Authors

Dear authors, 

Your manuscript has clinical relevance, using a diverse Reader Cohort. The results consistently show that AI improves diagnostic accuracy, particularly among less experienced readers, a key finding for workflow optimization in emergency settings.

However, your study uses a single expert thoracic radiologist as the reference standard. While experienced, interobserver validation with a second reader or consensus panel would enhance reliability.

Please check these small inconsistency issues: 

  1. Use standardized terms consistently. For instance, “front-line group” is somewhat vague; consider “junior readers” or specify exact roles consistently throughout.
  2. Describe in more detail what the AI presents to the reader (e.g., bounding boxes, heat maps, confidence score) and whether readers could adjust or interact with the AI output.
  3. The discussion around the potential for AI in training/education of junior radiographers or PGY residents could be expanded.
  4. Consider referencing DECIDE-AI or CONSORT-AI to ensure AI-specific aspects are thoroughly reported, especially regarding software versioning, training data, and calibration.

Good luck.

Author Response

Response to Reviewer 2 Comments

1. Summary

We sincerely thank Reviewer 2 and editor for their thoughtful and encouraging feedback. We truly appreciate their recognition of our study’s clinical relevance and their constructive comments that have guided us in refining both the clarity and completeness of our manuscript. In particular, the reviewer’s suggestions prompted us to (1) clarify and reinforce the reliability of our reference standard, (2) ensure consistency in terminology throughout the text, (3) elaborate on the AI interface and reader interaction process, and (4) expand our discussion on AI’s role in education and training for junior radiographers and PGY residents.

We also reviewed relevant AI reporting frameworks, including DECIDE-AI and CONSORT-AI, to confirm the adequacy of our methodological reporting and align it with best practices in AI-related research. We are grateful for Reviewer 2’s insightful input, which helped us recognize several aspects that required additional detail and precision, thereby improving the manuscript’s transparency, educational value, and overall scientific rigor. Detailed responses to each point are provided below.

2. Questions for General Evaluation

Reviewer’s Evaluation

Response and Revisions

Does the introduction provide sufficient background and include all relevant references?

Yes

Is the research design appropriate?

Yes

Are the methods adequately described?

Yes

Are the results clearly presented?

Can be improved

We have reanalyzed the data with updated statistical methods and presented the results in a more concise and structured manner. The revised Results section now focuses on key findings with clearer explanation of significant outcomes.

Are the conclusions supported by the results?

Can be improved

According to the new results, we revise our discussion, and some figures and tables have been reformatted for clarity.

Are all figures and tables clear and well-presented?

Can be improved

Several figures and tables that were previously unclear have been revised for improved clarity, and additional tables have been incorporated to more effectively illustrate and support our conclusions.

3. Point-by-point response to Comments and Suggestions for Authors

Comments 1: Dear authors, your manuscript has clinical relevance, using a diverse Reader Cohort. The results consistently show that AI improves diagnostic accuracy, particularly among less experienced readers, a key finding for workflow optimization in emergency settings. However, your study uses a single expert thoracic radiologist as the reference standard. While experienced, interobserver validation with a second reader or consensus panel would enhance reliability.

Response 1: Thank you for this important point. The radiologist used a serial CXR, CT scan and clinical chart to make the final diagnosis, not a single CXR. Therefore, the reference standard should be reliable. We have provided a more detailed description in the revised manuscript (page 3, Materials and Methods section, line 106-107).

Comments 2: Use standardized terms consistently. For instance, “front-line group” is somewhat vague; consider “junior readers” or specify exact roles consistently throughout.

Response 2: Thank you for highlighting this important point. We acknowledge that inconsistent terminology may cause confusion for readers. We have revised the term “front-line group” to “junior readers” in the abstract, as recommended (page 1, Abstract section, line 32). In addition, we have clarified the use of the term “front-line healthcare providers” throughout the manuscript to avoid potential ambiguity.

Comments 3: Describe in more detail what the AI presents to the reader (e.g., bounding boxes, heat maps, confidence score) and whether readers could adjust or interact with the AI output.

Response 3: The Lunit system provided heat maps and confidence score of each pattern, including Nodule, Fibrosis, Atelectasis, Cardiomegaly, Pneumothorax, Calcification, Consolidation, Pleural Effusion, Pneumoperitoneum, Mediastinal Widening on a second capture of CXR. Readers can not adjust the heat map but can adjust the score report on the RIS. We have provided a more detailed description in the revised manuscript (page 3, Materials and Methods section, line 84-87).

Comments 4: The discussion around the potential for AI in training/education of junior radiographers or PGY residents could be expanded.

Response 4: Thank you for this valuable suggestion. We agree that AI has the potential to play an important role in training readers. To address this, we have expanded our discussion section to include a paragraph highlighting how AI could be used as a supportive educational tool (page 11, Discussion section, line 288-295).

Comments 5: Consider referencing DECIDE-AI or CONSORT-AI to ensure AI-specific aspects are thoroughly reported, especially regarding software versioning, training data, and calibration.

Response 5: Thank you for this constructive suggestion. Since our study used a commercially available AI model rather than a self-developed algorithm, detailed information regarding software versioning, training dataset, and validation methodology was obtained from the publication by Nam et al. (“Development and validation of a deep learning algorithm detecting 10 common abnormalities on chest radiographs,” Eur Respir J, 2021). Their paper provides a comprehensive description of these aspects.

We also reviewed the DECIDE-AI and CONSORT-AI frameworks to ensure appropriate coverage of AI-related elements. While our study is not an RCT or model development research, most relevant AI-specific reporting standards were already addressed in Nam et al.’s validation study.

4. Response to Comments on the Quality of English Language

Point 1: The English is fine and does not require any improvement.

Response 1: Thank you for your positive feedback. We have carefully re-reviewed the entire manuscript and made English language revisions to improve readability and clarity for the readers.

5. Additional clarifications

NA

Reviewer 3 Report

Comments and Suggestions for Authors

Congratulations to you on your successful usage of AI assistance in pneumothorax interpretation for chest radiography.  The entire manuscript was well written in acceptable language and layout. I think your result will be of valuable contribution for the future clinical practice. I have only one comment and that is the'Discussion' is too lengthy and will be better if you can streamline.

Author Response

Response to Reviewer 3 Comments

1. Summary

We sincerely thank Reviewer 3 and the editor for their kind recognition of our work and encouraging comments. We are truly grateful for the reviewer’s positive feedback regarding the manuscript’s clarity, organization, and clinical value. Such affirmation has been a great source of motivation for our team.

In response to the reviewer’s helpful suggestion, we have carefully streamlined the Discussion section to improve focus and readability while preserving the key interpretations and clinical implications of our findings. We hope these revisions further enhance the overall quality and conciseness of the manuscript.

2. Questions for General Evaluation

Reviewer’s Evaluation

Response and Revisions

Does the introduction provide sufficient background and include all relevant references?

Yes

Is the research design appropriate?

Yes

Are the methods adequately described?

Yes

Are the results clearly presented?

Yes

Are the conclusions supported by the results?

Yes

Are all figures and tables clear and well-presented?

Can be improved

Several figures and tables that were previously unclear have been revised for improved clarity, and additional tables have been incorporated to more effectively illustrate and support our conclusions.

3. Point-by-point response to Comments and Suggestions for Authors

Comments 1: Congratulations to you on your successful usage of AI assistance in pneumothorax interpretation for chest radiography.  The entire manuscript was well written in acceptable language and layout. I think your result will be of valuable contribution for the future clinical practice. I have only one comment and that is the'Discussion' is too lengthy and will be better if you can streamline.

Response 1: Thank you for your precious opinion. We made efforts to concise our results and discussion when we revise our manuscript, and we hope it elevates the quality of our article.

4. Response to Comments on the Quality of English Language

Point 1: The English is fine and does not require any improvement.

Response 1: Thank you for your positive feedback. We have carefully re-reviewed the entire manuscript and made English language revisions to improve readability and clarity for the readers.

5. Additional clarifications

NA

Round 2

Reviewer 1 Report

Comments and Suggestions for Authors

The authors have revised the article in accordance with the review comments. Accept the article.